# Electro-Optical and Photo Stabilization Study of Nematic Ternary Mixture

**DOI:** 10.3390/ma14092283

**Published:** 2021-04-28

**Authors:** Aleksandra Kalbarczyk, Noureddine Bennis, Jakub Herman, Leszek R. Jaroszewicz, Przemysław Kula

**Affiliations:** Faculty of Advanced Technologies and Chemistry, Military University of Technology, 2 gen. S. Kaliskiego St., 00-908 Warsaw, Poland; aleksandra.kalbarczyk@wat.edu.pl (A.K.); jakub.herman@wat.edu.pl (J.H.); jarosz@wat.edu.pl (L.R.J.); przemyslaw.kula@wat.edu.pl (P.K.)

**Keywords:** frequency-controlled birefringence, photostability, response time

## Abstract

Liquid crystal materials composed of mixed nematic compounds find broad use in liquid crystal displays and photonic applications. A ternary mixture formed from three different nematic compounds shows peculiar behavior such as tunable electro-optical properties dependent on the frequency of the driving voltage. The paper presents an analysis of the response time and phase retardation of a frequency tunable nematic liquid crystal mixture (under code name 5005). This material possesses high birefringence (Δ*n* = 0.32 at 633 nm) as well as high dielectric anisotropy (Δ*ε* = 6.3 at 100 Hz). The unique property of the 5005 mixture is frequency-controlled phase modulation, as in a dual frequency liquid crystal, while dielectric anisotropy goes to zero instead of being negative at high frequencies. For each component of the mixture, details on mesomorphic properties and their role in the formulation of the mixture are reported. The 5005 mixture was characterized by multiple investigation techniques, such as temperature dependence dielectric anisotropy, transmittance measurements image polarizing microscopy, and UV stability.

## 1. Introduction

Liquid crystal’s (LC) optical properties are electrically tunable, and therefore they are attractive for displays [1,2,3,4,5,6,7,8], spatial light modulators [9,10,11], laser beam steering [11,12,13], lenses [14,15,16], filters [17,18], etc. A crucial parameter of LC materials is their dynamics. A voltage-controlled birefringence LC selectively modulates the polarization, amplitude and/or phase of an impinging wavefront. This kind of voltage control tends to cause fringing-field effects on high-resolution liquid crystal micro display devices [19]. This effect originates from different unequal voltage values applied to adjacent pixels, and the phase response between adjacent pixel electrodes can be affected [20]. Furthermore, due to the intrinsic anisotropy of nematic liquid crystals, the focusing properties of LC micro-lenses present astigmatism and aberration when a low voltage value is applied [21]. In our previous work, we have demonstrated that astigmatism and aberration can be minimized in LC micro-lenses with tunable focal length determined by frequency, keeping driving voltage constant [14]. Therefore, there is a need for LC material with electro-optical properties adjusted by the frequency of signal control for a given voltage. The optical characteristics of the frequency-controlled birefringence effect were observed in dual-frequency liquid crystals (DFLCs). In this case, the dielectric anisotropy of the LC changes the sign by switching the frequency of the applied voltage from below to above the crossover frequency. A study of the electro-optical properties of DFLCs shows that they may be suitable for applications that require fast response time [22]. On the other hand, for a given voltage, switching between two frequencies only allows us to obtain two switching states for the LC director. This is of importance particularly for the binary optical switch. Furthermore, when high frequencies are applied to larger heat dissipation, stronger dielectric heating is produced [23]. This limitation prevents dual-frequency nematic liquid crystals from finding commercial application. We would like to emphasize here that the multi-frequency LC studied in this work is different to conventional DFLCs; in fact, the dielectric anisotropy goes to zero instead of being negative at high frequencies [24]. The multi-frequency-controlled LC mixture presented in this work is a ternary mixture based on three LC components, two of which are dielectrically positive and one that is dielectrically neutral. The result is an LC material with large dielectric anisotropy that is continuously controlled by frequency. The dynamic dielectric behavior of such a mixture has been measured in the frequency range 100 Hz–10 kHz. In this work, we report the static dielectric measurement and its stability with temperature. It is predicted that the proposed mixture exhibits a high birefringence, promising to be used in photonic devices, and such devices require a large phase shift i.e., above 2π. This condition sometimes requires thicker cells, causing problems with high response time. To obtain large phase modulation in nematic LC (NLC), the cell gap can be increased, contributing to the slow response of the device. In recent years, scientists have been extensively working on high birefringence LCs [4,25,26,27,28,29,30]. The main disadvantages of these materials still remain their slow response time and, in most cases, instability when exposed to UV radiation.

To improve the response time of the LC device, manipulation with a driving signal, temperature, thickness of the LC layer or changes to the order of the molecules can be used. Furthermore, special materials have been designed to meet the requirements of fast response time, such as the cholesterics [31], blue phases [32] or very fast DFLCs [33,34] mentioned above. An interesting and promising method to speed up the response time is polymerization of the LC host with a monomer to obtain a polymer network LC (PNLC) and polymer-stabilized LC (PSLC) [35,36,37]. The process of photopolymerization using a UV light is a standard method to obtain polymer networks, significantly reducing the response time, with an important drawback; Polymer stabilized structures need a high driving voltage. Besides such a disadvantage, high birefringence LCs contain compounds with triple bonds or double bonds, which makes them unstable to UV radiation due to photochemical reactions, and this limits their use as hosts for LC in the photopolymerization process. The major problem of UV instability remains, even though methods to improve ultraviolet tolerance have been reported [38,39,40]. Although many secondary effects are associated with UV light, in this work, we have evaluated the UV stability of our mixture. Our results indicate reduction in the effective value of birefringence and a decrease in the threshold voltage.

In this paper, we study a high birefringence NLC material with Δ*n* = 0.32 and high dielectric anisotropy Δ*ε*, which is positive in the working frequency range. This mixture (named 5005) was formulated in the Institute of Chemistry at Military University of Technology in Warsaw, and it can be controlled similarly to DFLC, but its frequencies are low, and the sign of dielectric anisotropy does not change. Furthermore, using this mixture, multiple levels of phase modulation, tunable by adjusting the frequency of the applied signal from 3–34 kHz range, can be obtained [24]. This has a positive effect on the switch-ON time, which is strongly dependent on the applied voltage, when the amplitude modulation is applied. To study the relaxation dynamics, a temperature-dependent birefringence for different frequencies has been measured. We describe the influence of temperature on the dielectric properties, and UV stability tests have been performed.

## 2. Physical Properties of LC Host

High birefringence LC materials are mostly multicomponent mixtures. The final working composition should have adequate birefringence, dielectric anisotropy and chemical stability in a given temperature range of the nematic phase. The investigated NLC (5005 mixture) is a composition of three different families with rod-like molecular shapes. The chemical structures of components are presented in Figure 1. The components were selected so that the final mixture has a high birefringence, ∆*n* = 0.32 at 633 nm (at room temperature); wide nematic range; and positive high dielectric anisotropy at low frequencies (below 10 kHz).

The component 1 group are the fluorine substituted cyano-diester derivatives 4-[(4-cyanophenoxy) carbonyl] phenyl 4-alkylbenzoates with phase transition to isotropic liquid (*T_Iso_*) above 200 °C. It is a highly polar, dielectrically positive nematic material with very high dielectric anisotropy ∆*ε* (above 60 at 1 kHz) and a high dipole moment of 12.4 Debye. The substitution with fluorine (F) atoms lowers the threshold voltage, the ability to form a smectic phase, and the melting point. The cyano group -CN and neighboring ester groups -COO- have the greatest influence on the value of the dipole moment and nematogenity. The resulting dipole moment of the polar groups stays along the principal molecular axis, and therefore ∆*ε* is large. A larger ∆*ε* helps to reduce the switching voltage. However, the molecules of the group 1 compound itself are relatively long and generate high viscosity. The longer the molecule, the more viscous the LC medium, and hence the slower the molecule rotation and the lower its relaxation frequency. Dielectric anisotropy Δ*ε* is strongly dependent on the frequency of the applied voltage. At low frequencies, it has a positive sign, and at higher frequencies, it becomes negative due to the dispersion of *ε*_‖_ in the kilohertz range. Therefore, this family of materials has been used in the past as key components of DFLCs [25,26,41,42].

The component 2 group are alkyl-alkyl bistolanes with *T_Iso_* close to 150 °C. Due to the presence of lateral alkyl chains, they are strongly nematogenic. This group of materials was selected to behave as a highly birefringent LC solvent medium to minimize smectic phase formation. It is an almost dielectrically neutral and non-polar liquid crystal material. Since its dipole moment is close to zero, its ∆*ε* is very small. However, the triple bonds between benzene rings improve the conjugation of π electrons, and Δ*n* is expected to be very high [26,27,28,29,30].

The component 3 group are the fluorine substituted alkyl-alkyl phenyl-tolanes with phase transitions to *T_Iso_* at close to 90 °C. This group of compounds was selected to increase the miscibility between group 1 and 2 since they show common features of both groups (rigid core composed of tolane unity similar to group 2 and lateral fluoro-subsitution similar to group 1). Group 3 compounds are a dielectrically positive nematics, behaving similar to the compounds of group 1, with dielectric anisotropy dependent in range of kilohertz frequencies. Due to the presence of benzene rings connected via a triple bond, the birefringence is relatively high. The functionalization of molecules by the electronegative atom (F) enables the reorientation of molecules with the electric field and increases the value of ∆*ε* [29,30].

With the above-mentioned components, a nematic mixture with frequency-controlled birefringence at a range of frequencies (100 Hz—34 kHz) has been formulated as the 5005 mixture. As result of this combination, the LC mixture is a positive nematic with ∆*ε* = 6.3 at 100 Hz, birefringence ∆*n* = 0.32, and clearing temperature at about 120 °C. In fact, the results reported in [14,24] show that at room temperature, the electro-optical properties of this mixture are frequency tunable at kHz range. To measure the dielectric permittivity, the 5005 mixture has been injected in cells with gold electrodes and 0.7 mm thickness. The substrates were spin-coated by polyimide SE-130 (Nissan Chemical Industries), baked at 180 °C for 1 h and gently rubbed. Then, both substrates were assembled in antiparallel orientation. The experimental setup to measure the dielectric constant was similar to the one described in our previous work [24]. The samples were oriented by magnetic field to parallel and perpendicular orientation. The temperature dependence of the real part of the parallel *ε*_‖_ and perpendicular *ε*_⊥_ dielectric constant of component 2, component 3 and the 5005 mixture are presented in Figure 2. The values of Δ*ε* were taken at 100 Hz and at low voltages (below threshold value) for each sample. It should be emphasized that while component 3, which contributes to the formulation of the 5005 mixture, has lower clearing temperature, the dielectric properties of this mixture in the range between 25 and 60 °C have been measured where the three former components are present. For component 2 [Figure 2a], component 3 [Figure 2b] and the 5005 mixture [Figure 2c], the parallel constant of dielectric permittivity decreases gradually with temperature, while the perpendicular is kept almost constant. Hence, the dielectric response of the mixture should be the result of the response of individual components. Dielectric measurement of neat component 1 was rather difficult to carry out due to its high viscosity and very high melting point, which makes the characterization of the principal molecular motions in the order of seconds. In addition, their high ionic conductivity caused by free charges that are completely impossible to remove makes characterization very difficult [41], and therefore the dielectric properties of this compound have to be measured in other neutral nematic matrices in order to extrapolate their resulting dielectric properties. The temperature dependence of dielectric permittivity was evaluated from the results of a Demus’ ester mixture with 10% of compound 1 and presented in [42], assuming the additive contributions of each component to the permittivities. Compound 1 exhibits very high dielectric anisotropy at a wide range of temperatures due to large *ε*_‖_ caused by pronounced conductivity of the substance. Strong ionic conductivity in spite of several re-crystallization procedures could be caused by polarization due to free charges (the so-called Maxwell-Wagner effect) [41,43].

The results of Figure 2 show that material with non-polar molecules (component 2) has a small value of dielectric anisotropy due to the induced electronic and ionic polarization. In addition to induced electronic and ionic polarization in component 3, orientation polarization occurs due to the permanent dipole moment, and in fact this component has polar molecules groups and high anisotropy of molecular polarizability. The dielectric anisotropy of both components slowly decreases as temperature increases and drops to zero as the temperature approaches the clearing point, indicating a molecular disordering. The formulated mixture has larger dielectric anisotropy than component 2 and component 3 separately, which confirms that component 1 contributes significantly to the increase of the dielectric anisotropy of the 5005 mixture. The results in Figure 2c show that our mixture remains stable over the temperature range (20–65 °C).

Electro-optical characterizations of the mixture have been performed in 5 μm thick cells made with an indium-tin-oxide (ITO)-coated glass plate covered by a rubbed alignment layer and filled with 5005 mixture. The cell was mounted in a Linkam heating stage to control the temperature during the measurements. The optical retardation of the homogenously aligned LC cell sandwiched between two crossed linear polarizers was extrapolated from the VT curve. The retardance was calculated by using the equation $\varphi =2\arcsin\left(\sqrt{T_{Ʇ}}\right)$, where $T_{Ʇ}$ is the normalized transmission obtained for an LC cell oriented at 45° with respect to the polarizer orientation. This configuration was implemented between a He-Ne laser (*λ* = 633 nm) and a photodetector. The temperature dependence of the electro-optical performance in the range 25–115 °C was measured by applying a square-wave AC signal with different frequencies to the LC cell. Figure 3 shows the results of the temperature dependence of the VT curve at three frequencies (1 kHz, 10 kHz and 20 kHz). For an applied signal of a given frequency, the optical retardation uniformly decreases as the temperature increases. In addition, higher frequencies increase the threshold voltage and change the retardation modulation depth. On the other hand, the results show that the temperature change affects the voltage value at which the cell could develop a different phase level by changing the frequency of the applied voltage. Particularly at *T* = 25 °C, the biggest tuning of the optical retardation by frequency was obtained for 5.3 V (about 3.8π), while at *T* = 40 °C, the voltage was 3.29 V, which correspond to (about 2.1π) retardation in the frequency range (1 kHz–20 kHz). However, the range modulation depth of the phase shift by changing the frequency of the applied signal tends to decrease with increasing temperature. Figure 4a shows the relationship between the effect of the frequency and the temperature on optical retardation when the voltage applied to the cell is 10 V. At a temperature of 25 °C, the phase difference between 1 kHz and 20 kHz is about 1.7π, while above 55 °C it is practically small. These results are surprising since even at 55 °C, the dielectric anisotropy measured for 5005 mixture was high, ∆*ε* = 5.3. This could be explained by the fact that component 1 has the highest value of ∆*ε*, showing DFLC properties that can be strongly temperature dependent. The fact that an increase in the temperature shifts the crossover frequency of DFLC toward higher values [23,42] means that the frequency range of phase modulation tunability is reduced. For temperature values above *T_Iso_* of component 2, the optical response is independent to the frequency of the applied signal. This fact proves that the frequency dependence of the birefringence requires that all the three components need to be in the nematic phase.

Dynamic electro-optical characterizations of the cell were carried out to study the relaxation time of the mixture. The cell was addressed with a driving voltage 10 V square-wave AC signal modulated by a low frequency square wave. That is, the applied voltage to the LC was varied between 10 V and zero to switch the cell ON and OFF during the test. The measured rise time $\tau_{on}$ and decay time $\tau_{\mathrm{off}}$ are calculated between 90% and 10% of the maximum intensity change. The results of the relaxation times at different operating temperatures and for three different frequencies of the applied 10 V square AC signal (1 kHz, 10 kHz and 20 kHz) are plotted in Figure 4b. The response time *τ* presented in Figure 4b is the sum of $\tau_{on}$ and $\tau_{\mathrm{off}}$. At *T* = 25 °C, *τ* is equal to 130 ms, indicating a slow response of the LC mixture. As the temperature increases, the rotational viscosity of the mixture is reduced, and hence the response time tends to decrease. On the other hand, with an increase of the temperature up to 85 °C, the response time is less than 30 ms. Unlike voltage-driven LC cells, in the case of a frequency control scheme, the response time seems to be substantially independent of the applied frequency. This is another indication that the performance of devices where several levels of phase change are required with the same value of the response time can be beneficial for many applications. The dynamic response of this material can be improved by photo-polymerization, and therefore the study of stability of the LC mixture under UV irradiation should be considered.

## 3. UV Stability Test

Molecular structure and absorption play important roles in governing the UV stability of high birefringence LCs. Stability photochemistry in this spectral range is closely related to their spectra absorption, resulting from electronic transitions between different energy states. Electronic transitions using σ bond electrons (e.g., -CH_2_-CH_2_-) are related to the absorption of the shortest, more energetic waves in the range of 120–160 nm. Isolated CH = CH or C ≡ C bonds have absorption bands in the range of 160–200 nm. In turn, electronic transitions of the *n* → π * type and conjugated π → π * (e.g., -CH_2_O-, -C = O, -CH = CH-, -CH = CH_2_) correspond to the absorption of radiation from the longer ranges of 200–300 nm. For chemical systems with extensive π-electron conjugation, absorption is often shifted to even longer wavelengths (300—400 nm), which is caused by the reduction of the energy gap between the ground π and excited π * states of the molecules [44]. A double bond and carbon-carbon triple bond could initiate the photo-degradation process and then disturb LC alignment. As a result, the LC threshold voltage and effective birefringence are decreased. A UV stability test on the mixture has been performed to test the effect of tolane- and bistolane-based components in the investigated mixture. The 5005 mixture was filled into a 5-μm-thick LC cell with homogenous alignment thickness and ITO glass substrates. The experimental setup is the same as that for measuring the VT curves. The electro-optical response and pictures were taken for LC cell filled with 5005 material exposed to UV light (λ = 375 nm, power *P* = 40 mW/cm^2^) for time of exposure *t* equal to 60 s, 120 s, 5 min, 10 min, 60 min, 120 min, as well as a blue light (*λ* = 445 nm, *P* = 40 mW/cm^2^, *t* = 120 min). All tested cells have comparable thicknesses, listed in Table 1. Figure 5a shows the voltage-dependent transmittance curves, whereas Figure 5b shows the phase retardation for the fresh sample and the sample after UV and blue light exposure.

From Figure 5a, one can observe that with increasing time of exposure to UV light, light scattering is observed, and the threshold voltage is slightly decreased. The degradation mechanism is believed to originate from UV-induced free radicals. These free radicals transfer charges, converting triple bonds into double bonds and initiating the photodegradation process from front surface layers and gradually migrating into bulk as UV dosage increases [39]. Once the surface layers are cross-linked, the surface alignment is disturbed, resulting in a reduced Δ*n* [Figure 5b] and increased viscosity, smeared threshold voltage, or increased light scattering.

The performed UV stability test, including the observed degradation caused by UV light, have been documented by polarizing microscopy. The LC cells after exposure to UV radiation were mounted between crossed polarizers with an optical axis rotated 45° in respect to the polarizer orientation. Figure 6 presents the polarized optical micrographs images of 5005 mixture taken by the CCD camera mounted into the microscope. In Figure 6a, an LC cell filled with 5005 mixture is presented. From those results, we would like to emphasize that while the cell being exposed to UV radiation does not show degradation, it would require at least 60 s of irradiation time to trigger degradation. In the next pictures, Figure 6c–h, one can notice the degradation process of the material caused by UV light. After 60 s of irradiation time, formation of the droplets can be related to the change in chemical structure, especially triple-bond breaking in tolane and bistolane compounds. Furthermore, point and linear defects are formed, indicating discontinuities in the nematic order and disturbing the alignment of the liquid crystal molecules inside the cell.

We have attempted to evaluate the stability of our material using a blue laser at a wavelength 445 nm. Figure 7 indicates that after 2 h irradiation time, no damage has been found to the texture of the cells. This is of importance for using this material in applications such as projection displays.

High values of optical anisotropy can be generated by using combinations of aromatic rings and linking groups in the form of carbon-carbon multiple bonds for the construction of the molecular core. For compounds with such cores, the number of possible combinations (assuming the use of only three benzene rings and one or two acetylene bridges) is sufficient to obtain compounds with sufficient high birefringence for many potential applications. Unfortunately, the introduction of multiple bonds to the molecules greatly reduces the stability of such a structure in the UV area [40].

The degradation of LC materials under UV light is manifested by:(1)color change;(2)reducing the effective value of birefringence Δ*n*;(3)reduction and blurring of the threshold voltage; and(4)intensification of the light scattering process.

According to the work of Lin et al. [40], the starting point for this degradation is the free radicals of acetylene groups -C≡C- produced by UV radiation, which in turn leads to the initiation of the polymerization process. When the layers of molecules near the surface become cross-linked, the order of the molecules is disturbed. As a result, this leads to a reduction in the effective value of birefringence and an increase in viscosity, which causes an extension of the switching times and a decrease in the threshold voltage. The effect of the progressive degradation under UV of 5005 mixture components is noticeable in Figure 5 and Figure 6. Nematic 5005 mixture is composed of structures containing acetylene groups -C ≡ C- (either one for phenyl tolanes or both for bistolanes)—see Figure 8.

Among the liquid crystal structures with two ethynyl groups that are known so far, bistolanes are characterized as the most UV stable. Despite the fact that structures contain two ethynyl groups, -C≡C-, their separation with one aromatic ring can somehow lead to a slowing down of the degradation process under the influence of UV light. Taking into consideration the investigated 5005 mixture, bistolane derivatives are classified as the least UV stable. Nevertheless, it is also known that tolane structures are, among the described ones, the most resistant to UV radiation. This is due to the presence of only one ethynyl group, the reactivity of which, however, is still the weakest link in highly birefringent LC structures.

## 4. Conclusions

In this work, a ternary mixture formed by three different nematic components has been characterized by multiple investigation techniques, such as temperature dependence dielectric anisotropy, transmittance measurements, and image polarizing microscopy. The components were selected that the final mixture has high a birefringence, Δ*n* = 0.32 (at 633 nm at room temperature); wide nematic range; and positive high dielectric anisotropy at low frequencies (below 10 kHz). For each component of the mixture, details of their mesomorphic properties and their role in the formulation of the mixture are reported. Optical transmission measurements were performed to check the dependency of optical retardation on temperature and frequency. Under a longer period of UV irradiation, a degradation process has been observed in the microscope textures, and the VT characteristics shows birefringence and threshold voltage reduction and light scattering. However, irradiation of the material with a blue laser causes no damage to the texture of the cells. This UV stability study has proved that the investigated material cannot be exposed to ultraviolet radiation.

## Figures and Tables

**Figure 1 materials-14-02283-f001:**
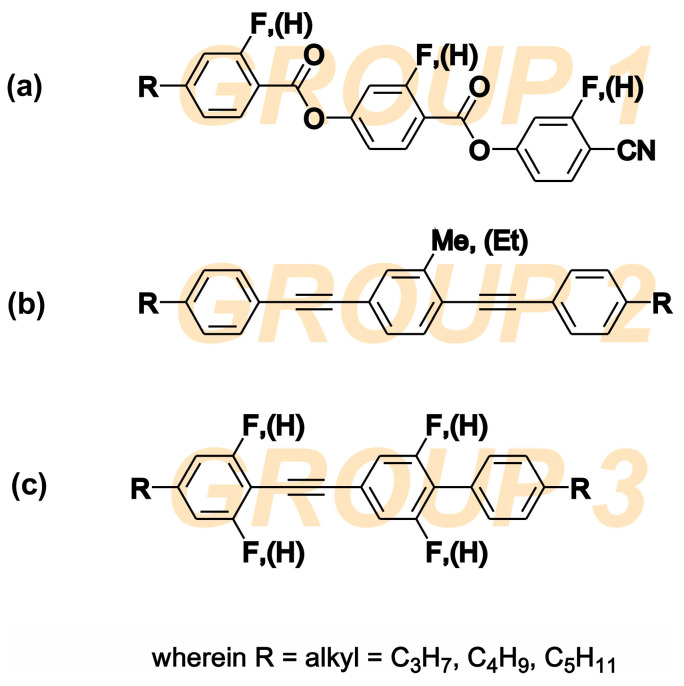
Chemical structures of three LC families combined to form the 5005 nematic mixture: (**a**) fluorine substituted 4-[(4-cyanophenoxy) carbonyl] phenyl 4-alkylbenzoates, (**b**) alkyl-alkyl bistolanes, and (**c**) fluorine substituted alkyl-alkyl phenyl-tolanes.

**Figure 2 materials-14-02283-f002:**
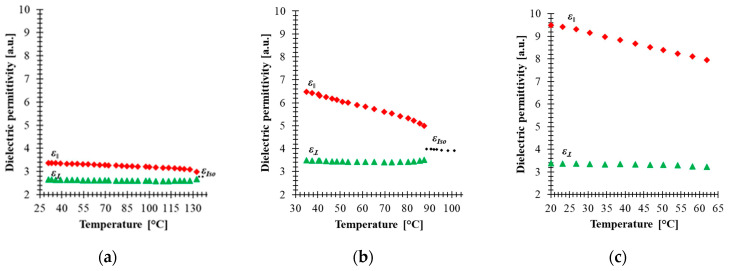
Temperature dependence of the real part of permittivity of: (**a**) component 2, (**b**) 3 and (**c**) 5005 mixture for 100 Hz. Red points—parallel constant of permittivity *ε*_‖_, green points—perpendicular constant of permittivity *ε*_⊥_, black points—permittivity in the isotropic state *ε_Iso_*.

**Figure 3 materials-14-02283-f003:**
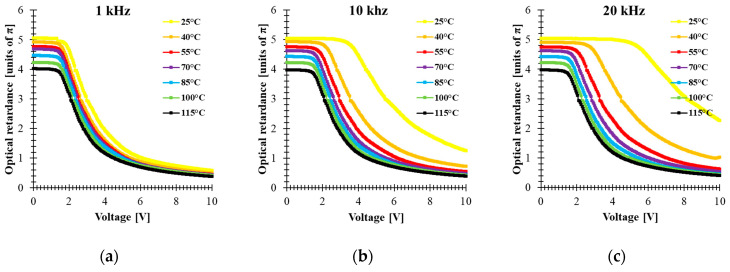
Voltage-dependent optical retardance of 5005 mixture at different temperatures (*λ* = 633 nm, cell gap 5 μm) for different frequencies of applied square AC signal: (**a**) 1 kHz, (**b**) 10 kHz, (**c**) 20 kHz.

**Figure 4 materials-14-02283-f004:**
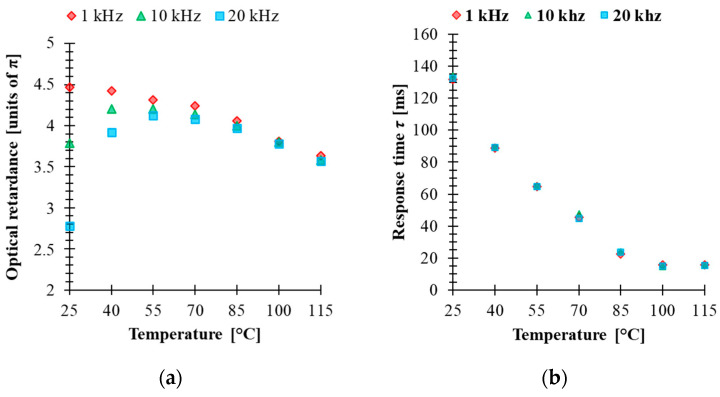
(**a**) Temperature-dependent optical retardance and (**b**) temperature-dependent response time of 5005 mixture (*λ* = 633 nm, cell gap 5 μm) while controlled by 1 kHz, 10 kHz and 20 kHz of 10 V square-wave AC signal.

**Figure 5 materials-14-02283-f005:**
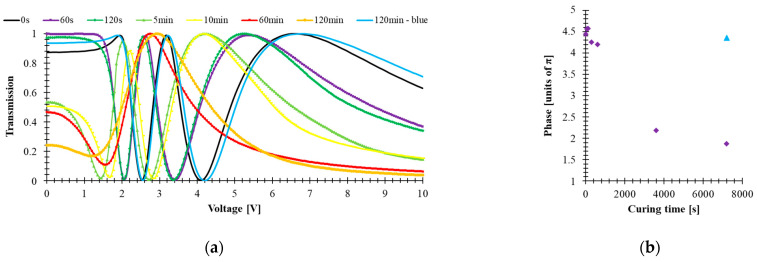
(**a**) Voltage-dependent transmittance and (**b**) phase change of 5005 mixture under UV (time of exposure: 60 s, 120 s, 5 min, 10 min, 60 min, 120 min, power 40 mW/cm^2^, *λ* = 385 nm) and blue light (time of exposure 120 min, power 40 mW/cm^2^, *λ* = 445 nm) exposition.

**Figure 6 materials-14-02283-f006:**
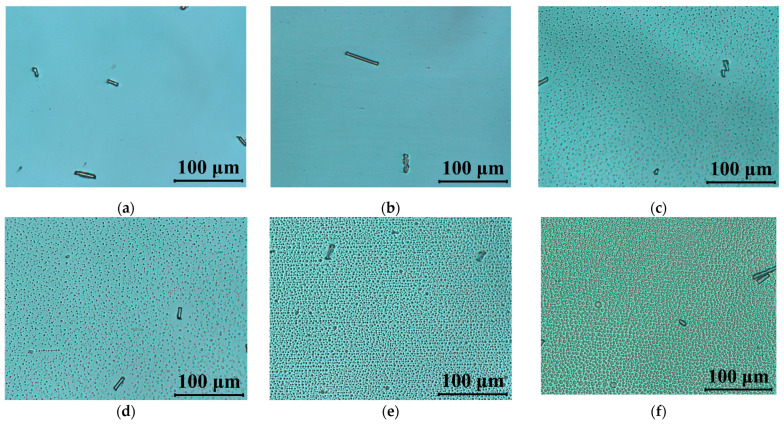
Polarized optical micrographs of cells filled with 5005 mixture observed between crossed polarizers: (**a**) fresh sample; and (**b**–**h**) under UV light (*λ* = 375 nm, *P* = 40 mW/cm^2^) during: (**b**) 10 s, (**c**) 60 s, (**d**) 120 s, (**e**) 5 min, (**f**) 10 min, (**g**) 60 min, (**h**) 120 min. The LC’s optical axis was rotated at 45° with respect to the polarizer orientation.

**Figure 7 materials-14-02283-f007:**
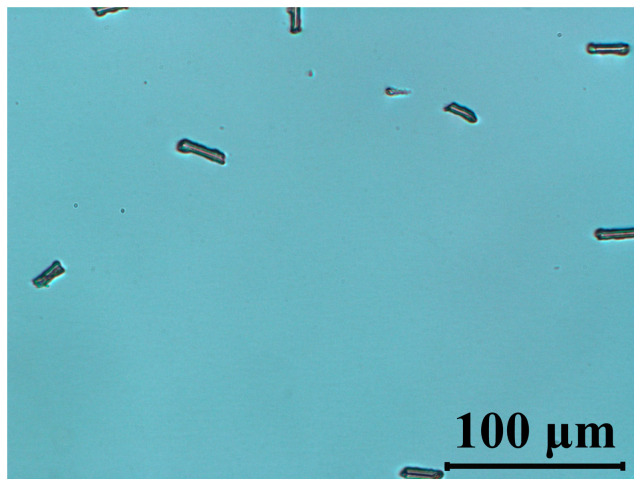
Polarized optical micrographs of cells filled with 5005 mixture observed between crossed polarizers under blue light (*λ* = 445 nm, *P* = 40 mW/cm^2^, *t* =120 min). The LC’s optical axis was rotated at 45° with respect to polarizer orientation.

**Figure 8 materials-14-02283-f008:**
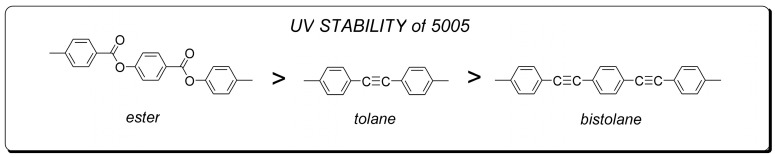
Comparison of UV stability of 5005 mixture ingredients.

**Table 1 materials-14-02283-t001:** Thicknesses of LC cells for UV and blue light stability test.

Photoirradiation Time	Cell Thickness
0	5.13 μm
10 s	5.07 μm
60 s	5.13 μm
120 s	5.12 μm
5 min	5.14 μm
10 min	5.16 μm
60 min	5.13 μm
120 min	5.09 μm
120 min–blue light	5.13 μm

## Data Availability

Not applicable.

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
