# Peer review of "Electro-Optical and Photo Stabilization Study of Nematic Ternary Mixture"

_materials, 2021, doi:10.3390/ma14092283_

Round 1
Reviewer 1 Report
I attached the reviewer's report.

Author Response
We would like to thank for the time and effort invested in the review of our manuscript, and for helpful comments and suggestions. We revised the manuscript with special attention to all received comments and we believe that the Reviewer’s suggestions have improved our manuscript. In the following, we have addressed all comments as can be seen in the enclosed list. The original Reviewer’s comments are shown in italics.
REVIEWER 1
In this paper, the authors addressed verifying the frequency tunability of optical retardance and UV stability of a liquid crystal mixture composed of three high birefringence nematics. Those properties are important for liquid crystal displays, lenses, and other technologies. The authors measured the optical retardance in the long-range region, kHz. The obtained results are good, and this paper is basically well-written. I think this paper reaches the criteria of the journal Materials, and recommend it to be published. However, there are several issues and careless mistakes, for which I commented below, before publication. The authors also should check the whole again by themselves.
- I recommend you to provide the ε‖, ε⊥, and Δε of component 1 in Figure 2 to visually catch the role of component 1 in those the mixture 5005 shown in Figure 2 (c). In addition, you mentioned as below.
“These results are surprising since even at 55 °C the dielectric anisotropy measured for 5005 mixture was high Δε = 5.3. This could be explained by the fact that component 1 has the highest value of Δε, showing DFLC properties that can be strongly temperature dependent.”
Response: Thank you for your suggestion. We agree that the results for component 1 would make a big contribution to the article. The problem of measuring the temperature dependence of dielectric permittivity in component 1 had been described in our work. Due to very high viscosity and high dielectric anisotropy, the measurements have to be performed using a solution of component 1 and a mixture having small negative dielectric anisotropy (Demus’ ester mixture). The results extrapolated from the mixture Base 903 and 10% of compound 1, assuming the additive contributions of both component to the permittivities have been reported in [a] – substance 5. We have implemented the description of the results in Section 2, lines 165-173.
[a] Czub, J; Urban, S.; Ziobro, D.; DÄ…browski, R. Dielectric Properties of Strongly Polar Nematogens with Cyano and Fluoro Substituents. Mol. Cryst. Liq. Cryst. 2009, 506, 286-295; DOI: 10.1080/15421400903065762.
- What is the role of the following phrase ? Because, the component 2 (TIso = 90 °C) contributes to lower the TIso not component 3 (TIso = 150 °C).
“It should be emphasized that while a component which contribute to the formulation of the 5005 mixture has lower clearing temperature, the dielectric properties of this mixture in the range between 25 and 60 °C have been measured, where the three former components are present.”
Response: Thank you for your observation. The graphs of the temperature dependence of permittivity for compound 2 and 3 have been swapped. We have changed Figure 2 and corrected mistakes on 3rd page. We apologize for the mistake.
- I’m confused in Figure 2 panels (a) and (b) and related sentences. The caption of Figure 2 says panel (a) and (b) denote components 2 and 3, respectively. In the main text, however, you mentioned about component 2 as below:
It is a dielectrically neutral and non-polar liquid crystal material. Since its dipole moment is zero, thus its Δε is very small. The results of Figure 2 show that material with non-polar molecules (component 2) has small value of dielectric anisotropy, due to the induced electronic and ionic polarization. In addition to induced electronic and ionic polarization in component 3, an orientation polarization occurs due to the permanent dipole moment, in fact this component has polar molecules groups and high anisotropy of molecular polarizability.
Response: Thank you for your remark. As mentioned above, in Figure 2, panels (a) and (b) were swapped. We have corrected our mistake.
Nevertheless, panel (a) exhibits clearly larger ε‖, ε⊥, and Δε than panel (b). In addition, you say the component 2 has zero or neutral Δε in the earlier part, but it is indeed not zero.
Response: Component 2 is indeed not exactly dielectrically neutral. Its Δε is very small comparing to other components of the 5005 (but still > 0) and therefore our expression in the manuscript was not sufficiently correct. We have implemented the corrections to the decription of component 2 (lines 126-127).
- It seems that Figures 6 (h) and 7 are the same.
Response: We apologize for the mistake. Figure 6(h) have been changed to the correct one.
- A preceding work reported that UV degradation caused an increase threshold voltage due to an increase in viscosity. But, your result indicated that UV degradation caused threshold voltage reduction?
Response: In ref. [38] the blurring and decrease in threshold voltage was observed, while the degradation occurs. In our paper we have also observed this phenomenon (Fig. 5). We have revised the text carefully and found the mistake in the text, line 308. It has been corrected.
Thank you for all the suggestions and comments. They indeed greatly improve the readability of our manuscript.
Reviewer 2 Report
The authors characterize a novel LC mixture (5005) composed of three components. The material shows high birefringence and dielectric anisotropy. The material has a frequency-dependent response to voltage, which makes it suitable for applications. The UV-instability is very typical for high birefringence material, and the 5005 mixture is no exception. However, the authors state that the mixture's stability is good for blue light, which is essential for possible applications.
Overall I find the paper to be well written. However, some issues stated below prevent me from accepting the article in its present form. Please address these questions before resubmitting the paper.
1.) How did you determine the optical retardance? Did you use a commercial device, or did you reconstruct the phase from the TV curve? For this one usually needs a numerical model Fredericksz transition model. Can you explain more in the text, please?
2.) Figures 6 and 7 must have a scale bar. Also, please specify the analyzer and polarizer's orientations and LC alignment direction in the captions or draw them in the images.
3.) Figure 6.h) is not correct. You must have mistaken it with Figure 7. Please add the right image here.
Below are some remarks that authors may address:
a.) I was slightly puzzled by the 5005 naming. At first, I got the impression that this is a standard material that one can buy. After reading the text, it got clear that it is just your "code name." It may be better to tell the reader (in the last paragraph of the introduction) that you prepared the mixture and track it under your internal code name.
Author Response
We would like to thank for the time and effort invested in the review of our manuscript, and for helpful comments and suggestions. We revised the manuscript with special attention to all received comments and we believe that the Reviewer’s suggestions have improved our manuscript. In the following, we have addressed all comments as can be seen in the enclosed list. The original Reviewer’s comments are shown in italics.
The authors characterize a novel LC mixture (5005) composed of three components. The material shows high birefringence and dielectric anisotropy. The material has a frequency-dependent response to voltage, which makes it suitable for applications. The UV-instability is very typical for high birefringence material, and the 5005 mixture is no exception. However, the authors state that the mixture's stability is good for blue light, which is essential for possible applications.
Overall I find the paper to be well written. However, some issues stated below prevent me from accepting the article in its present form. Please address these questions before resubmitting the paper.
- How did you determine the optical retardance? Did you use a commercial device, or did you reconstruct the phase from the VT curve? For this one usually needs a numerical model Fredericksz transition model. Can you explain more in the text, please?
Response: The optical retardation was extrapolated from VT curve, by using the equation . This information was implemented to the text in the manuscript (lines 191-196).
- Figures 6 and 7 must have a scale bar. Also, please specify the analyzer and polarizer's orientations and LC alignment direction in the captions or draw them in the images.
Response: Thank you for your comment. The scale bars have been added to Figures 6 and 7 and the orientations of polarizers and LC have been described in the text (line 280), as well as in the headings of Figures 6 and 7.
- Figure 6.h) is not correct. You must have mistaken it with Figure 7. Please add the right image here.
Response: We apologize for the mistake. Figure 6(h) have been changed to the correct one.
Below are some remarks that authors may address:
- I was slightly puzzled by the 5005 naming. At first, I got the impression that this is a standard material that one can buy. After reading the text, it got clear that it is just your "code name." It may be better to tell the reader (in the last paragraph of the introduction) that you prepared the mixture and track it under your internal code name.
Response: Thank you for your suggestion. This description was written at the begging of Section 2. We have implemented this information at the last paragraph of the introduction – see lines 83-84.
Thank you for all the suggestions and comments. They indeed greatly improve the readability of our manuscript.
Reviewer 3 Report
The article is generally well written, even though there are some typos, and the presentation is clear and coherent. I recommend its publication in materials after minor revision as follows. The authors are required to address the following comments and suggestions: i) specify the electrodes in the experiments to measure the dielectric permittivity; ii) in fig. 2 (c) report the data for all the nematic range and some points in the isotropic phase, as in graphs (a) and (b) of the same figure; iii) the high dielectric anisotropy observed in mixture 5005 could be induced by the ionic effect of component 1? Discuss this point; iv) provide high-resolution images.
Author Response
We would like to thank for the time and effort invested in the review of our manuscript, and for helpful comments and suggestions. We revised the manuscript with special attention to all received comments and we believe that the Reviewer’s suggestions have improved our manuscript. In the following, we have addressed all comments as can be seen in the enclosed list. The original Reviewer’s comments are shown in italics.
The article is generally well written, even though there are some typos, and the presentation is clear and coherent. I recommend its publication in materials after minor revision as follows. The authors are required to address the following comments and suggestions:
- specify the electrodes in the experiments to measure the dielectric permittivity;
Response: The electrodes to measure the dielectric permittivity were gold. We have added this information in line 146 of the manuscript.
- in fig. 2 (c) report the data for all the nematic range and some points in the isotropic phase, as in graphs (a) and (b) of the same figure;
Response: The dielectric temperature dependence of the nematic material 5005 has been determined up to approximately 65 °C, due to the fact that this multicomponent mixture is considered by us as an applicable material. Therefore, the measurement was performed for a typical temperature range in which an electro-optical device using liquid crystal material operates.
- the high dielectric anisotropy observed in mixture 5005 could be induced by the ionic effect of component 1? Discuss this point;
Response: Thank you for asking that. Component 1 has very high Δε. We believe that it has very high value of ε‖ (above 200), due to high ionic conductivity. Because the substance shows a large ionic conductivity in spite of several re-crystallization procedures, it was assumed that the enormously large ε‖ was caused by the polarization due to free charges (so called Maxwell-Wagner effect [a]). In addition, the concentration of compound 1 in the mixture is below 10%, but it certainly increases the value of Δε in 5005. Extrapolated results of dependence of dielectric permittivities on temperature change for component 1 have been reported in [b] – substance 5. We have implemented the description of the results in Section 2, lines 165-173.
[a] Daniel, V. V. (1967). Dielectric Relaxation, Academic Press: New York.
[b] Czub, J; Urban, S.; Ziobro, D.; DÄ…browski, R. Dielectric Properties of Strongly Polar Nematogens with Cyano and Fluoro Substituents. Mol. Cryst. Liq. Cryst. 2009, 506, 286-295
- provide high-resolution images.
Response: We have improved the resolution of the Figures 6 and 7 and provided a scale bars on the photographs.
Thank you for all the suggestions and comments. They indeed greatly improve the readability of our manuscript.